# Growth patterns of children under 5 years old in rural area northern of Abha, Aseer Region, Saudi Arabia

**Safar Abadi Alsaleem** *

Department of Family and Community Medicine, King Khalid University, Abha, Saudi Arabia

* asalslim@kku.edu.sa

## Abstract

### Introduction

Measuring and monitoring physical growth is a key element in assessing children's health. The physical and mental development of a growing child are indicators of good health and nutrition. This study aimed to explore the patterns of growth parameters among children under 5 years old attending primary health care centres (PHCCs) in rural northern Abha. Feeding habits during the first 2 years of life were also assessed.

### Method

This study was conducted on 1–5 April 2022 in all (PHCCs) in Billasamr and Ballahmar, two neighbourhoods of northern Abha. The investigator designed a form for collecting each child's age (months), weight (kilograms), gender, birth order, number of family members, type of feeding (exclusive breastfeeding, exclusive bottle feeding, or mixed feeding), and duration of exclusive breastfeeding.

### Results

Gender affected the body weight of Saudi infants, as boys weighed significantly more than age-matched girls. Birth order also affected body weight. During the first and third years of life, Saudi children's body weight was not significantly affected by feeding type. During the second and fifth years of life, formula-fed children weighed more than breastfed or combination-fed children. However, during the fourth year of life, body weight was highest in combination-fed children.

### Conclusion

Growth patterns over time, established from multiple data points, must be used in conjunction with other medical and family history to evaluate appropriate growth. Recording and tracking body size, especially recumbent length and weight, as part of well-baby care is critical for any assessment of child wellbeing and health status.

**Data Availability Statement:** All relevant data are within the paper and its Supporting Information files

**Funding:** The author(s) received no specific funding for this work.

**Competing interests:** On behalf of of myself, disclose any competing interests that could be perceived to bias this work—acknowledging all financial support and any other relevant financial or non-financial competing interests.

**Abbreviations:** PHCC, primary health care centre; WHO, world health organization; km, kilometre; ANOVA, analysis of variance; SPSS, Statistical Package for the Social Sciences; MRT, Duncan's new multiple range test; p-value, probability value; KSA, Kingdom of Saudi Arabia; kg, kilogram; Min, minimum; Max, maximum.

## Introduction

Growth is defined as an increase in body size via the proliferation of cells and the production of intercellular components over a period of time. Growth is also considered a form of motion, and human growth is closely related to sufficient nutrition that contains adequate calories to meet developmentally specific needs. Growth is a sensitive predictor of social health [1, 2].

Measures of children's physical growth are considered key indicators of health. Growth and mental development are indicators of good health and nutrition in children. Growth curves are constructed as a reference for screening and monitoring patterns of growth, although not as a criterion for definite malnutrition or pathology [3]. Child growth and development are multifactorial issues with environmental, socioeconomic, and biological aspects. The monitoring of child growth is necessary, as deficits in these parameters can have negative effects [4, 5]. The most effective way to determine child growth is by measuring birth weight and then tracking weight gain over time [2].

In 1978, the WHO identified primary health care (PHC) as central to achieving the goal of "Health for All" and as a key instrument for improving health worldwide [6]. PHC was adopted as a basic mechanism to promote health care to the population and is delivered through the District Health System because it was the most accessible and cost-effective means to improve the health status of the population. This system brings healthcare as close as possible to the public and constitutes the first element of the continuing health care process [7, 8].

The provision of high-quality health care requires knowledge of normal patterns that help distinguish health from disease. Many countries have established their own comprehensive growth charts for their children and adolescents. For example, in the Kingdom of Saudi Arabia (KSA), attempts have been made to assess the growth of children [9].

Abha, the capital city of the Aseer Region, is located at an altitude of 3133 m above sea level in the mountains of southwestern Saudi Arabia. Previous studies have shown that term infants in Abha are lighter and shorter than age-matched Saudi infants from the surrounding low-altitude areas [10].

This study was carried out to explore the pattern of growth parameters among children under 5 years old attending primary health care centres (PHCCs) in the rural area of northern Abha. Feeding habits during the first 2 years of life were also assessed.

## Materials and methods

### Study setting and population

This study was conducted after obtaining approval from The Research Ethics Committee at King Khalid University (ECM#2023–3302) and written consent obtained from parents and guardians of the minors included in the study during the period from the 1st to the 5th of April 2022 at all PHCCs in the Billasmar and Ballahmar neighbourhoods of northern Abha. The city of Abha is located 3133 m above sea level in the Aseer Mountains of southwestern Saudi Arabia. The northern area of Abha is located 50 km away from the city centre.

Billasmar and Ballahmar contain a total of 14 PHCCs (Fig 1) [11], through which all health services, including well-baby care and general practice, are offered to all citizens free of charge. Well-baby care and services are provided by well-trained nurses and general practitioner doctors on national recommendations issued by the Ministry of Health. Each PHCC center provides health services to 13,455 people, on an average [12].

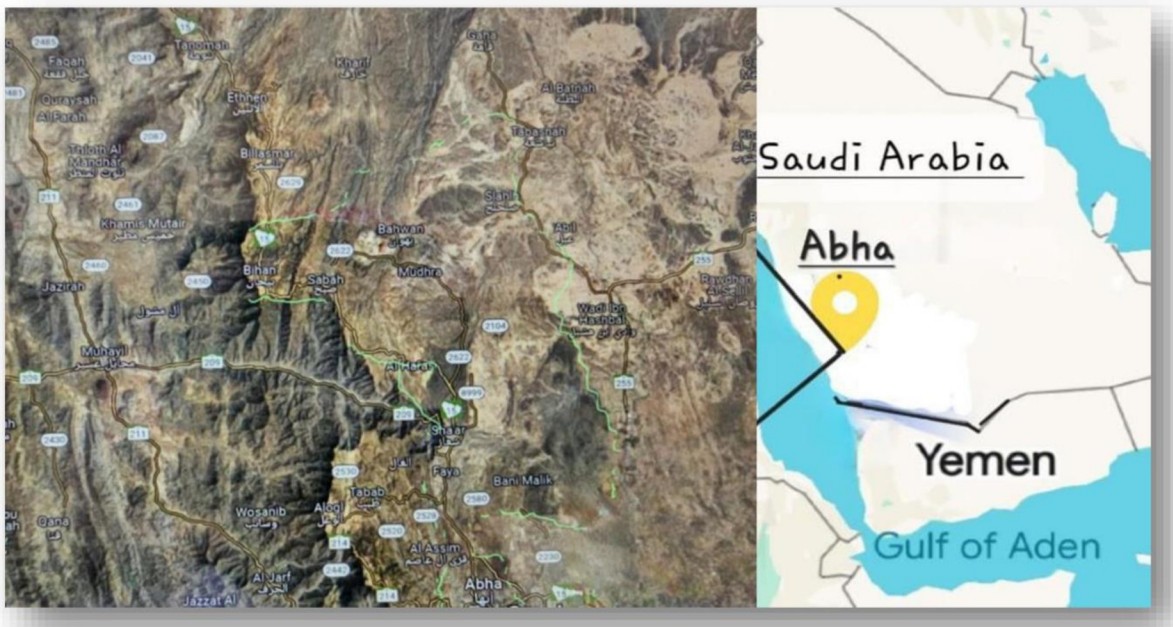

**Fig 1.** Fourteen Billasamr and Ballahmar PHCs: 1-Sabah 2-Mawin 3-bahwan 4-Slahlh 5-Ebil 6-Shada 7-Abalah 8-Bihan 9-Billasmar 10-Alhussin 11-Hawra 12-Kharif 13-Sadwan 14-Almdfah. (Reference: https://www.google.com.sa/maps/@18.6301018,42.5326289,101076m/data=!3m1!1e3?hl= en&entry=ttu last accessed 01 Dec 2023).

### Data collection

Data were collected during Health Campaign 2022, which lasted one week. The investigator designed a collection sheet to record all relevant data regarding children aged 5 years and younger who attended the clinic. The sheet included the gender, birth order, and type of feeding (exclusive breastfeeding, exclusive bottle feeding, or mixed feeding)

Working nurses in the primary health care centres were given written instructions regarding the objective and importance of this study and how to measure weight and fill out the data collection sheet properly. After one week, all data collection sheets from all centres were collected by the principal investigator.

**Operational definitions.** *Gender.* Gender is a categorical variable that categorizes children into two groups: "boys" and "girls." The classification is based on the child's biological sex, with boys being individuals assigned as male at birth and girls as individuals assigned as female at birth.

*Birth order.* Birth order is a categorical variable that assigns each child to one of the following categories: "1st," "2nd," "3rd," or "4th," based on their position in the order of birth within their family. It is determined by asking parents about their children's birth order.

*Type of diet (Feeding).* The type of diet or feeding is a categorical variable that classifies the child's diet during their early years. It includes three categories: "Breast feeding, Artificial feeding and Mixed feeding," This categorization is based on the primary source of nutrition the child receives.

*Sample size.* It was determined using the formula, $n = z^2 p(1-p)/d^2$, where, p = 50.0% using Z (1.96) at 95% confidence interval, margin of error (3.56%), the sample size was calculated to

be 730. We employed an appropriate sample size using 95% confidence interval and 80% statistical power.

## Statistical analysis

Data were coded and then entered into a personal computer to be analysed by ANOVA in the IBM software program SPSS (version 16.0; IBM Corp., NY, USA) The differences between means were tested for significance using Duncan's new multiple range test (MRT). A difference was considered significant if its p-value was <0.05. The PROC REG procedure was performed to evaluate the dependency of the body weight of Saudi children in their first five years on the duration of breastfeeding.

Descriptive statistics primarily focused on three key factors: Gender, Order of child, and Type of feeding. The tables represent important statistics, including the mean value for each category, expressed as "Mean ± SE," with SE denoting the standard error representing precision. Additionally, the table provides the minimum (Min) and maximum (Max) values within each category, indicating the lowest and highest recorded values for the respective variable. These statistics aid in summarizing data distributions, central tendency, and variability within each category.

In statistical analysis, ANOVA has been applied to examine differences in means across various groups within the factors of Order of child, and Type of feeding. For instance, in the "The order of child" and "Type of feeding" factors, ANOVA is applied to assess mean differences between different birth orders and types of feeding, in the "Gender" factor, an independent t test is used to determine whether there are significant differences in body weight between boys and girls.

The IQR (Interquartile Range) method for identifying and potentially removing outliers involves calculating the IQR, which is the range between the 25th and 75th percentiles of the data. Outliers are removed from the dataset, and retained for further analysis.

## Results

The total number of participants in the study is 730.

Table 1 displays the frequency distribution of birth order and gender among Children aged one to five years. These demographic patterns, suggest potential variations in gender distribution based on birth order during specific age intervals. In the first year, the 2nd order has the highest percentage of boys (63.60%), while the 1st order has the highest percentage of girls (61.80%). Similar patterns persist in the second and third years, with a relatively balanced gender distribution across birth order categories. However, in the fourth year, a notable shift is observed, with the 1st order having the lowest percentage of boys (29.70%) and the highest percentage of girls (70.30%). In the fifth year, the ≥4th order stands out with the highest percentage of boys (62.10%), while the 3rd order has the highest percentage of girls (53.30%).

Table 2 presents the frequency distribution of feeding practices about gender among children aged one to five years. These findings highlight potential diversity in feeding practices in gender-specific preferences in certain age groups. In the first year, a balanced distribution is observed across genders for exclusive breastfeeding and mixed feeding, while artificial feeding shows a slight preference for boys (50.00%). In the second year, a higher percentage of boys are fed with artificial and mixed feeds, while exclusive breastfeeding is more evenly distributed. The third year continues this pattern, with a slight inclination towards exclusive breastfeeding for girls. In the fourth year, there is a preference for girls in exclusive breastfeeding and boys in artificial feeding. The fifth-year exhibits a similar trend, with boys having a higher percentage of artificial feeding.

**Table 1. Age and gender-wise distribution of children according to birth order.**

| Birth Order of the child | 1st | | 2nd | | 3rd | | ≥4th | |
|---|---|---|---|---|---|---|---|---|
| Gender | Boy | Girl | Boy | Girl | Boy | Girl | Boy | Girl |
| The first year of age | 13 | 21 | 21 | 12 | 14 | 9 | 22 | 21 |
| | 38.20% | 61.80% | 63.60% | 36.40% | 60.90% | 39.10% | 51.20% | 48.80% |
| Second year of age | 23 | 21 | 16 | 18 | 9 | 14 | 19 | 17 |
| | 52.30% | 47.70% | 47.10% | 52.90% | 39.10% | 60.90% | 52.80% | 47.20% |
| The third year of age | 19 | 16 | 18 | 17 | 16 | 24 | 28 | 27 |
| | 54.30% | 45.70% | 51.40% | 48.60% | 40.00% | 60.00% | 50.90% | 49.10% |
| The fourth year of age | 11 | 26 | 14 | 11 | 14 | 11 | 23 | 22 |
| | 29.70% | 70.30% | 56.00% | 44.00% | 56.00% | 44.00% | 51.10% | 48.90% |
| Fifth year of age | 22 | 19 | 19 | 15 | 14 | 16 | 36 | 22 |
| | 53.70% | 46.30% | 55.90% | 44.10% | 46.70% | 53.30% | 62.10% | 37.90% |

Table 3 shows that there was no significant difference in body weight between the sexes during the first year of life; the recorded mean values (kg) were 9.40± 0.26 and 9.58±0.15 for boys and girls, respectively. Additionally, infants who were 3rd in birth order had the highest mean body weight (kg), 10.20± 0.12, followed by 1st (9.77 ± 0.30) and 2nd children (9.68 ± 0.25). Moreover, during the 1st year of life, there was no significant difference in body weight among Saudi infants receiving different diets (breast milk, formula, or both). The mean values of body weight (kg) were 9.55±0.32, 9.75±0.16 and 9.45±0.18 for children who consumed breast milk, formula and a combination, respectively (Table 1).

Concerning the body weight of Saudi infants during the second year of life, Table 4 reveals that body weight (kg) was significantly affected by gender, birth order and type of diet. Boys had higher body weight (11.52 ± 0.23 kg) than girls (10.75 ± 0.21 kg). Additionally, formula feeding resulted in a significantly higher body weight (13.05 ± 0.41 kg) than breastfeeding (10.99 ± 0.29 kg) or mixed feeding (10.87 ± 0.19). Meanwhile, 1st and 4th children had slightly highest body weight (11.48 ± 0.31 kg and 11.49 ± 0.14 kg, respectively) 2nd or 3rd children (10.26 ± 0.42 kg and 10.97 ± 0.29 kg, respectively).

Table 5 shows that the body weight (kg) of Saudi children was affected by gender during the third year of life. Boys had significantly higher body weight (13.74 ± 0.17 kg) than girls (13.14 ± 0.15 kg). Moreover, children who were 3rd in birth order had the highest mean body weight, at 13.82±0.29 kg, but all differences in weight by birth order were nonsignificant, with

**Table 2. Age and gender-wise distribution of children according to type of feeding.**

| Type of feeding | Exclusive breastfeeding | | Artificial feeding | | Mixed feeding | |
|---|---|---|---|---|---|---|
| Gender | Boy | Girl | Boy | Girl | Boy | Girl |
| The first year of age | 33 | 38 | 2 | 2 | 35 | 23 |
| | 46.50% | 53.50% | 50.00% | 50.00% | 60.30% | 39.70% |
| Second year of age | 19 | 22 | 6 | 3 | 42 | 45 |
| | 46.30% | 53.70% | 66.70% | 33.30% | 48.30% | 51.70% |
| The third year of age | 27 | 34 | 6 | 5 | 48 | 45 |
| | 44.30% | 55.70% | 54.50% | 45.50% | 51.60% | 48.40% |
| The fourth year of age | 22 | 27 | 5 | 7 | 35 | 36 |
| | 44.90% | 55.10% | 41.70% | 58.30% | 49.30% | 50.70% |
| Fifth year of age | 22 | 24 | 8 | 8 | 61 | 40 |
| | 47.80% | 52.20% | 50.00% | 50.00% | 60.40% | 39.60% |

**Table 3. Factor affecting growth of Saudi infants during the first year of age.**

| Factors | Class | Mean ±SE | Min | Max |
|---|---|---|---|---|
| Gender | | | | |
| | Boys | 9.40±0.26 | 7.50 | 11.30 |
| | Girls | 9.58±0.15 | 8.20 | 10.90 |
| Order of child | | | | |
| | 1st | 9.77±0.30 [a] | 6.70 | 12.00 |
| | 2nd | 9.68±0.25 [a] | 8.50 | 11.30 |
| | 3rd | 10.20±0.12 [a] | 10.00 | 10.30 |
| | ≥4th | 8.83±0.19 [b] | 7.80 | 10.50 |
| Type of feeding | | | | |
| | Breast feed | 9.55±0.32 | 7.90 | 12.00 |
| | Artificial feed | 9.75±0.16 | 9.50 | 10.00 |
| | Mixed feed | 9.45±0.18 | 6.70 | 11.30 |

• Means with different superscripts in each category are significantly different at $p < 0.05$

1st, 2nd and 4th children weighing 13.19±0.23, 13.50±0.18 and 13.06±0.21 kg, respectively. Regarding types of feeding, there was no significant weight difference among children who had been breastfed (13.36±0.18 kg), those who had been formula fed (13.44±0.290 kg), and those who had been combination fed (13.34±0.16 kg).

Table 6 shows that, during the 4th year of life, body weight (kg) was significantly affected by gender and type of feeding. Interestingly, boys still had significantly higher body weight (16.22 ± 0.28 kg) than girls (15.40 ± 0.19 kg). Additionally, both breastfeeding and mixed feeding resulted in significantly higher body weight (16.50± 0.32 kg and 15.69 ± 0.23 kg, respectively) than formula feeding (14.20 ± 0.35 kg).

Table 7 showed that there was no significant weight difference between the sexes during the fifth year of life; the mean recorded values (kg) were 17.15± 0.28 and 16.60±0.33 for boys and girls, respectively. Children who were 4th in birth order had the highest body weight (kg), with a mean value of 17.37± 0.38, followed by 3rd (16.92 ± 0.51) and 1st children (16.75 ± 0.35). Moreover, during the fifth year of life, the body weight of Saudi children was significantly

**Table 4. Factor affecting growth of Saudi infants during the second year of age.**

| Factors | Class | Mean ±SE | Min | Max |
|---|---|---|---|---|
| Gender | | | | |
| | Boys | 11.52±0.23 [a] | 9.50 | 13.00 |
| | Girls | 10.75±0.21 [b] | 8.90 | 12.50 |
| Order of child | | | | |
| | 1st | 11.48±0.31 [a] | 9.00 | 15.7 |
| | 2nd | 10.26±0.42 [b] | 9.50 | 14.00 |
| | 3rd | 10.97±0.29 [b] | 8.50 | 12.50 |
| | ≥4th | 11.49±0.14 [a] | 9.50 | 13.00 |
| Type of feeding | | | | |
| | Breast feed | 10.99±0.29 [b] | 9.50 | 13.50 |
| | Artificial feed | 13.05±0.41 [a] | 11.50 | 13.00 |
| | Mixed feed | 10.87±0.19 [b] | 8.50 | 12.00 |

• Means with different superscripts in each category are significantly different at $p < 0.05$

**Table 5. Factor affecting growth of Saudi infants during the third year of age.**

| Factors | Class | Mean ±SE | Min | Max |
|---|---|---|---|---|
| Gender | | | | |
| | Boys | 13.74±0.17 [a] | 11.00 | 15.00 |
| | Girls | 13.14±0.15 [b] | 12.30 | 14.50 |
| Order of child | | | | |
| | 1st | 13.19±0.23 | 10.50 | 15.30 |
| | 2nd | 13.50±0.18 | 11.80 | 15.00 |
| | 3rd | 13.82±0.29 | 12.60 | 16.00 |
| | ≥4th | 13.06±0.21 | 11.00 | 14.50 |
| Type of feeding | | | | |
| | Breast feed | 13.36±0.18 | 12.00 | 16.60 |
| | Artificial feed | 13.44±0.29 | 10.00 | 15.00 |
| | Mixed feed | 13.34±0.16 | 10.50 | 15.30 |

• Means with different superscripts in each category are significantly different at $p<0.05$

affected by the type of diet they had received as infants. Children who had been exclusively breastfed or formula fed (17.69 ± 0.47 kg and 17.32. ± 0.39 kg, respectively) weighed more than those who had received both breast milk and formula (16.35 ± 0.32 kg).

Fig 2 presents the regression equation of the significant ($p < 0.01$) correlation between the body weight of Saudi infants and the duration of breastfeeding, indicating that the body weight of Saudi children in the 5th year of life is heavily dependent on the duration of breastfeeding.

## Discussion

The results presented in Table 1 revealed that there was no significant difference in body weight between boys and girls during the first year of life. Additionally, infants' body weight (kg) was significantly affected by birth order, increasing until the 3rd child (the mean values were 9.77 ± 0.30, 9.68 ± 0.25 and 10.20± 0.12 kg in 1st, 2nd and 3rd children, respectively) and then declined thereafter (8.83± 0.19). These results are consistent with a previous report [1] showing that birth order has a significant effect on weight. In that study, first and second children had a

**Table 6. Factor affecting growth of Saudi infants during the fourth year of age.**

| Factors | Class | Mean ±SE | Min | Max |
|---|---|---|---|---|
| Gender | | | | |
| | Boys | 16.22±0.28 [a] | 13.50 | 19.50 |
| | Girls | 15.40±0.19 [b] | 13.00 | 18.50 |
| Order of child | | | | |
| | 1st | 16.22±0.31 | 13.50 | 20.00 |
| | 2nd | 15.13±0.22 | 12.80 | 17.00 |
| | 3rd | 15.78±0.33 | 14.00 | 19.50 |
| | ≥4th | 15.74±0.32 | 13.00 | 20.00 |
| Type of feeding | | | | |
| | Breast feed | 16.50±0.32 [a] | 14.20 | 20.00 |
| | Artificial feed | 14.20±0.35 [b] | 12.80 | 18.00 |
| | Mixed feed | 15.69±0.23 [a] | 13.00 | 20.00 |

• Means with different superscripts in each category are significantly different at $p<0.05$

Table 7. Factor affecting growth of Saudi infants during the fifth year of age.

| Factors | Class | Mean ±SE | Min | Max |
|---|---|---|---|---|
| Gender | | | | |
| | Boys | 17.15±0.28 | 16.00 | 23.00 |
| | Girls | 16.60±0.33 | 13.50 | 21.00 |
| Order of child | | | | |
| | 1st | 16.75±0.35 | 15.00 | 20.00 |
| | 2nd | 15.70±0.42 | 12.50 | 18.00 |
| | 3rd | 16.92±0.51 | 13.50 | 21.00 |
| | ≥4th | 17.37±0.38 | 12.50 | 20.00 |
| Type of feeding | | | | |
| | Breast feed | 17.69±0.47 [a] | 16.00 | 21.00 |
| | Artificial feed | 17.32±0.39 [ab] | 15.00 | 19.00 |
| | Mixed feed | 16.35±0.32 [b] | 12.50 | 20.00 |

• Means with different superscripts in each category are significantly different at $p < 0.05$

weight advantage over babies who came later in the birth order (4th and above). Another study found an association between low weight and birth order from the 7th child onward in Nigerian families, but there was no corresponding relationship earlier in the birth order [13]. Additionally, **Table 1** illustrates that there was no significant difference (i.e., no difference for which $p < 0.05$) in the mean body weight of Saudi infants in the 1st year of life depending on the type of feeding they received (Table 1). In a previous study performed in 2012, no significant relationship was found between breastfeeding duration and body weight [1].

The body weight (kg) of Saudi infants during the second year of age was significantly affected by gender, birth order and type of feeding (Table 2). Boys had significantly higher body weight (11.52 ± 0.23 kg) than girls (10.75 ± 0.21 kg). These results are consistent with a previous report [14] showing that boys were heavier than girls from birth to 5 years of age. Another study [15] also found that girls were lighter than boys. Moreover, two additional studies [16, 17] found that boys were heavier than girls from birth to 2 years of age. In the present study, formula feeding resulted in a significantly higher mean body weight than breastfeeding or mixed feeding. Additionally, the 1st and 4th children in the family weighed slightly more than the 2nd and 3rd children. It was found in one study that children who had been breastfed for a longer duration had significantly higher mean weight up to 5 years of age [16]. However, another study reported a non-significant relationship between breastfeeding duration and body weight [1].

During the third year of life, there was a significant gender-based difference ($p < 0.05$) in the mean body weight of Saudi children (**Table 3**). Specifically, boys had a significantly higher mean body weight (13.74 ± 0.17 kg) than girls (13.14 ± 0.15 kg). Very similar results were previously recorded in two other studies [14, 15], which also found that girls were lighter than boys. Moreover, **Table 3** shows that the highest mean body weight (kg) for any birth order was 13.82±0.29 for 3rd children, but this value was not significantly different from the mean weights of 1st, 2nd and 4th children (13.19±0.23, 13.50±0.18 and 13.06±0.21, respectively). Concerning types of feeding, there were no significant differences among breastfeeding, formula feeding and mixed feeding. These results are consistent with a previous study [1] showing that there was no significant relationship between breastfeeding duration and body weight.

Concerning the body weight of Saudi infants during the fourth year of life, **Table 4** showed that the body weight (kg) was significantly affected by both gender and type of feeding.

**Y = 9.0136 + 0.2770 X\*\***

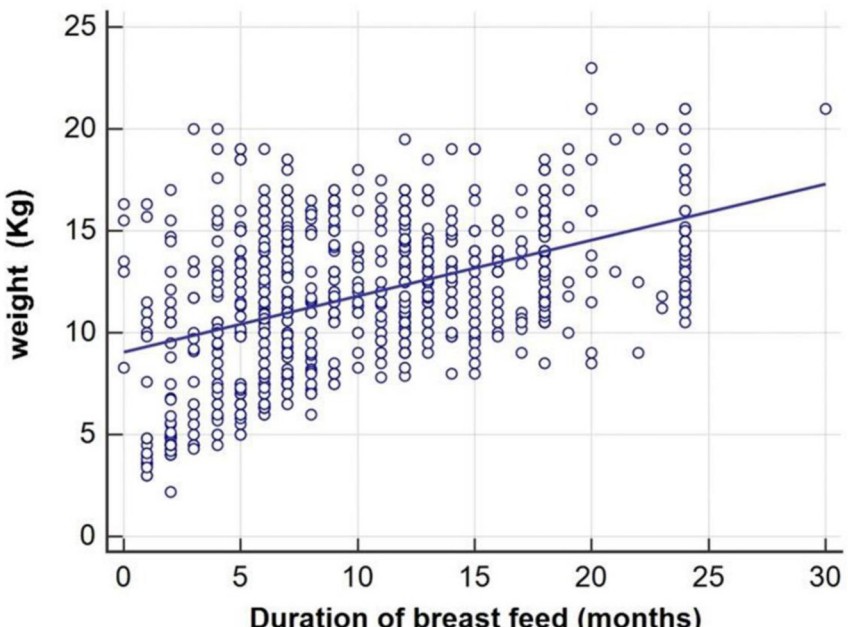

**Fig 2. The dependency of body weight (Kg) up to the fifth year of age on the duration of breast feed (months).**

Interestingly, boys still showed a significantly higher body weight ($16.22 \pm 0.28$) when compared to girls ($15.40 \pm 0.19$). Closely similar results were previously reported [14, 15] which showed that boys were heavier than girls from birth up to 5 years of age. Both breast feed and mixed feed resulted in significantly higher body weight ($16.50 \pm 0.32$ and $15.69 \pm 0.23$) compared to artificial feed ($14.20. \pm 0.35$). These results agreed with those cited before [14] who found infant's breastfed had a significantly higher mean weight, up to 5 years of age. While our results, disagreed with [1] who revealed a non-significant relationship between breastfeeding and body weight.

The body weight of Saudi infants during the fifth year of age was significantly affected by the type of feeding (**Table 5**). Interestingly, the breastfeed had resulted in a significantly higher body weight ($17.69 \pm 0.47$) compared to the mixed feed ($16.35 \pm 0.32$). Meanwhile, no significant variation was reported between the breast feed and artificial feed. These results were accepted by those previously reported [16] who found that infants breastfed for a longer duration had a significantly higher body weight.

**Fig 2** clarified the positive significant regression equation at level ($p < 0.01$) between the body weight of Saudi infants and duration of breast feed, indicating that the body weight of Saudi infants up to fifth year of age is greatly dependent on the duration of breast feed. These

findings did not agree with those of [1] who reported that there was a non-significant relationship between breastfeeding duration and body weight.

From above mentioned results, it was clear that the body weight of Saudi infants was affected by gender as boys showed a significantly higher body weight when compared to girls. Genes on the Y chromosome drive testosterone production both in the foetal stage and during the first few months of age. The increasing size and the growth rate of the male might be due to the production of testosterone [2]. Moreover, birth order has been shown to influence body weight. This phenomenon may occur because the 1st and 2nd child not only receive more attention than younger siblings but also face less competition for food [1]. Breastfeeding has many health benefits for infants, as breast milk meets all the nutritional needs of an infant and protects against common childhood illnesses; furthermore, breastfeeding may have longer-term health benefits for children and adolescents. Studies in developing countries have shown that continued, frequent breastfeeding is associated with increased linear growth and protects children's health.

While our study primarily focuses on the northern area of Abha city centre in Billasamr and Ballahmar which contains a total of 14 PHCCs, it's important to acknowledge that national-level studies offer a more comprehensive overview and typically involve larger sample sizes and broader geographic representation. To enhance the generalizability of our findings, we recommend future research to replicate our study in different areas or with diverse populations. The effect of weather in the study region should also be taken into consideration in future studies. Height was not considered due to difficulties in measuring especially length among infants. Unlike weight, accurate length measurement requires two people, one to help with positioning the child and the other to take the readings. At the level of the measurer/anthropometrist, monitoring the length of children can be challenging due to the vulnerability of length measurements to errors [18]. We hope in the future to have all the required resources such as equipment and training of health personnel to do height monitoring.

## Conclusion

This study was conducted as a reference study assessing child growth, and it established a strong basis for comparison with present and future measurements of Saudi children in the Aseer Region. These data can also be used in monitoring and promoting infant growth. Growth patterns over time, established from multiple data points, must be used in conjunction with other medical and family history to evaluate appropriate growth. Recording and tracking body size, especially recumbent length and weight, as part of well-baby care is critical for any assessment of a child wellbeing and health status.

## Supporting information

**S1 File.**
(DOCX)

## Author Contributions

**Conceptualization:** Safar Abadi Alsaleem.

**Data curation:** Safar Abadi Alsaleem.

**Formal analysis:** Safar Abadi Alsaleem.

**Methodology:** Safar Abadi Alsaleem.

**Resources:** Safar Abadi Alsaleem.

**Validation:** Safar Abadi Alsaleem.

**Visualization:** Safar Abadi Alsaleem.

**Writing – original draft:** Safar Abadi Alsaleem.

**Writing – review & editing:** Safar Abadi Alsaleem.

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
