## [Decision Letter · Decision Letter 0]

2 Nov 2023

PONE-D-23-32367Growth patterns of children under 5 years old in Rural area northern of Abha, Aseer Region, Saudi ArabiaPLOS ONE

Dear Dr. Alsaleem,

Thank you for submitting your manuscript to PLOS ONE. After careful consideration, we feel that it has merit but does not fully meet PLOS ONE’s publication criteria as it currently stands. Therefore, we invite you to submit a revised version of the manuscript that addresses the points raised during the review process.

We look forward to receiving your revised manuscript.

Kind regards,

Satish Rojekar, Ph.D.

Academic Editor

PLOS ONE

Journal Requirements:

Additional Editor Comments:

Dear Dr. Safar Abadi Alsaleem,

After carefully considering the reviewer's comments and my assessments, I recommend that the manuscript be thoroughly revised and resubmitted for further consideration.

Reviewer#1

In this article the author has investigated the effect of nutrition on the development of the child, where weight of the child was used as a growth marker. The study was carried out in the Bellasmar and Balhmer regions in Saudi Arabia.

I would like to ask author to work on the following points and to add the following information to the manuscript to help understand the data in coherent way.

1. General demographics of the area – population size, age group segregation, etc.

2. What is the average population size each PHCC serves to?

3. Total number of participants in the study?

4. Descriptive statistics of the whole data and sub-groups

5. Is ANOVA the best method to analyze the data?

6. Please add the criterion used for the selection and removal of the outliers.

7. I would request the authors to add the questionnaire as a supplementary data.

8. How do the findings of this study compare to other wider studies carried out on the national level.

9. What was the statistical power of the analysis and can the findings of this analysis be generalized to other areas?

Reviwer#2

This paper describes growth pattern for children under 5 years of age in rural abha region. It also included feeding habits as one of the criteria for evaluation. Gender, family history, medical history etc. combinedly provide the growth pattern over the time. The paper contains a large volume of data collection and statistical analysis, and is generally structured well, however, a number of points need to be justified before it can be accepted for publication.

General Comments:

• The manuscript contains a large number of grammar and formatting errors. Please have the manuscript proofread by a native English speaker.

• Graphical abstract shall be provided.

• Please verify the referencing style is per the journal guideline.

Specific Comments:

Introduction:

• The effect of weather in the study region should also be taken into consideration.

Method:

Study setting:

• Provide the approval number given by ethical committee for the study.

• A map showing the location of the PHCCs shall be added.

Data collection:

• Why is height not considered as one parameter?

Reviewers' comments:

Reviewer's Responses to Questions

**Comments to the Author**

1. Is the manuscript technically sound, and do the data support the conclusions?

Reviewer #1: Partly

Reviewer #2: Yes

2. Has the statistical analysis been performed appropriately and rigorously? 

Reviewer #1: No

Reviewer #2: Yes

3. Have the authors made all data underlying the findings in their manuscript fully available?

Reviewer #1: Yes

Reviewer #2: Yes

4. Is the manuscript presented in an intelligible fashion and written in standard English?

Reviewer #1: Yes

Reviewer #2: Yes

5. Review Comments to the Author

Reviewer #1: In this article the author has investigated the effect of nutrition on the development of the child, where weight of the child was used as a growth marker. The study was carried out in the Bellasmar and Balhmer regions in Saudi Arabia.

I would like to ask author to work on the following points and to add the following information to the manuscript to help understand the data in coherent way.

1. General demographics of the area – population size, age group segregation, etc.

2. What is the average population size each PHCC serves to?

3. Total number of participants in the study?

4. Descriptive statistics of the whole data and sub-groups

5. Is ANOVA the best method to analyze the data?

6. Please add the criterion used for the selection and removal of the outliers.

7. I would request the authors to add the questionnaire as a supplementary data.

8. How do the findings of this study compare to other wider studies carried out on the national level.

9. What was the statistical power of the analysis and can the findings of this analysis be generalized to other areas?

Reviewer #2: PONE-D-23-32367

Growth patterns of children under 5 years old in Rural area northern of Abha, Aseer Region, Saudi Arabia

This paper describes growth pattern for children under 5 years of age in rural abha region. It also included feeding habits as one of the criteria for evaluation. Gender, family history, medical history etc. combinedly provide the growth pattern over the time. The paper contains a large volume of data collection and statistical analysis, and is generally structured well, however, a number of points need to be justified before it can be accepted for publication.

General Comments:

• The manuscript contains a large number of grammar and formatting errors. Please have the manuscript proofread by a native English speaker.

• Graphical abstract shall be provided.

• Please verify the referencing style is per the journal guideline.

Specific Comments:

Introduction:

• The effect of weather in the study region should also be taken into consideration.

Method:

Study setting:

• Provide the approval number given by ethical committee for the study.

• A map showing the location of the PHCCs shall be added.

Data collection:

• Why is height not considered as one parameter?

6. PLOS authors have the option to publish the peer review history of their article (what does this mean?). If published, this will include your full peer review and any attached files.

Reviewer #1: No

Reviewer #2: No

---

## [Author Response · Author response to Decision Letter 0]

15 Dec 2023

Rebuttal letter: Response to the respected Academic Editor and respected Reviewers

As suggested by the respected Editor, Data Availability statement and Ethical Approval statements have been mentioned.

Reviewer 1

1. General demographics of the area – population size, age group segregation, etc.

Reply: As suggested by the respected reviewer the following has been added

Sample Size: It was determined using the formula, n=z^2 p(1-p)/d^2, where, p= 50.0% using Z (1.96) at 95% confidence interval, margin of error (3.56%), sample size was calculated to be 730. 

 (Please see Page 5, lines 27-30)

For age group segregation tables 1 and 2 have been added. (Please see page 17 and 18)

The following Operational definitions of variables have been added (page 5 lines 15-25)

Gender: Gender is a categorical variable that categorizes children into two groups: "boys" and "girls." The classification is based on the child's biological sex, with boys being individuals assigned as male at birth and girls as individuals assigned as female at birth.

Birth Order: Birth order is a categorical variable that assigns each child to one of the following categories: "1st," "2nd," "3rd," or "4th," based on their position in the order of birth within their family. It is determined by asking parents about their children's birth order.

Type of Diet (Feeding): The type of diet or feeding is a categorical variable that classifies the child's diet during their early years. It includes three categories: "Breast feeding, Artificial feeding and Mixed feeding," This categorization is based on the primary source of nutrition the child receives.

2. What is the average population size each PHCC serves to?

Reply:

Each PHC center provides health services to 13,455 people, on average. (Please see Page 5 lines 2-3) (https://www.moh.gov.sa/en/Ministry/Statistics/book/Documents/1433.pdf)

3. Total number of participants in the study?

Reply: The total number of participants in the study are 730. (page 7 line 6)

4. Descriptive statistics of the whole data and sub-groups

Reply: Under descriptive statistics, primarily focused on three key factors: Gender, Order of child, and Type of feeding. The table represent important statistics, including the mean value for each category, expressed as "Mean ± SE," with SE denoting the standard error representing precision. Additionally, the table provides the minimum (Min) and maximum (Max) values within each category, indicating the lowest and highest recorded values for the respective variable. These statistics aid in summarizing data distributions, central tendency, and variability within each category. (page 6 lines 10-16)

5. Is ANOVA the best method to analyze the data?

Reply: In statistical analysis, ANOVA has been applied to examine differences in means across various groups within the factors of Order of child, and Type of feeding. For instance, in the "Order of child" and "Type of feeding" factors, ANOVA is applied to assess mean differences between different birth orders and types of feeding, in the "Gender" factor, independent t test is used to determine whether there are significant differences in body weight between boys and girls. (page 6 lines 17-21)

6. Please add the criterion used for the selection and removal of the outliers.

Reply: The IQR (Interquartile Range) method for identifying and potentially removing outliers involves calculating the IQR, which is the range between the 25th and 75th percentiles of the data. Outliers is removed from the dataset, retained for further analysis. (page 6 lines 22-24)

7. I would request the authors to add the questionnaire as a supplementary data.

Reply: Thank you for your feedback regarding the questionnaire as supplementary data. We appreciate your comment. We have taken measures to ensure the privacy and confidentiality of the survey respondents, which prevented us from sharing the questionnaire directly. However, we would be more than willing to provide a detailed description of the questionnaire's structure, questions, and methodology used in our research, to help readers better understand the data collection process and the context of our study. We are open to further discussing this and addressing any specific questions or concerns you may have. (Questionnaire has been attached)

8. How do the findings of this study compare to other wider studies carried out on the national level.

Reply: Thank you for your question regarding the comparison of our study's findings to wider national-level studies. We appreciate the opportunity to contextualize our research within the broader landscape. While our study primarily focuses on the northern area of Abha city centre in Bellasmar and Balhmer which contain a total of 14 PHCCs, it’s important to acknowledge that national-level studies offer a more comprehensive overview and typically involve larger sample sizes and broader geographic representation. In our research, we aimed to address only the pattern of growth parameters among children under 5 years old those who attending primary health care centres (PHHCs), which may not be fully captured by national-level studies. (Please see page 12 lines 11-17)

9. What was the statistical power of the analysis and can the findings of this analysis be generalized to other areas?

Reply: Our analysis was conducted with careful attention to statistical power. We employed appropriate sample sizes using 95% confidence interval and 80% statistical power. We are glad we can explain our research in the bigger picture. While our study primarily focuses on the northern area of Abha city centre in Bellasmar and Balhmer which contain a total of 14 PHCCs, it's important to acknowledge that national-level studies offer a more comprehensive overview and typically involve larger sample sizes and broader geographic representation. To enhance the generalizability of our findings, we recommend future research to replicate our study in different areas or with diverse populations. (page 12 lines 11-17)

Reviwer#2

This paper describes growth pattern for children under 5 years of age in rural abha region. It also included feeding habits as one of the criteria for evaluation. Gender, family history, medical history etc. combinedly provide the growth pattern over the time. The paper contains a large volume of data collection and statistical analysis, and is generally structured well, however, a number of points need to be justified before it can be accepted for publication.

General Comments:

• The manuscript contains a large number of grammar and formatting errors. Please have the manuscript proofread by a native English speaker.

Reply: As suggested the manuscript has been proofread by a native English speaker.

• Graphical abstract shall be provided.

Reply: We need to work on this and would request the concerned journal team to help us with this. 

• Please verify the referencing style is per the journal guideline.

Reply: References have been revised as per the journals style.

Specific Comments:

Introduction:

• The effect of weather in the study region should also be taken into consideration.

Reply: As pointed, we have not taken this parameter and now included as a study limitation. Page 12 lines 16-17

Method:

Study setting:

• Provide the approval number given by ethical committee for the study.

Reply: Approval number has been added. (Page 4 line 19)

• A map showing the location of the PHCCs shall be added ( Figure ?????) .

Reply: As suggested we have added a map showing the location of the PHCCs. 

Data collection:

• Why is height not considered as one parameter?

Reply: Height was not considered due to difficulties in measuring especially length among infants. Unlike weight, accurate length measurement requires two people, one to help with positioning the child and the other to take the readings. At the level of the measurer/anthropometrist, monitoring length of children can be challenging due to the vulnerability of length measurements to errors. We hope in the future to have all the required resources such as equipment and training of health personnel to do height monitoring. We have mentioned this as a study limitation. (Please see Page 12 lines 17-22)

Thank you for helping us to improve our manuscript based on the valuable editorial and review c

---

## [Decision Letter · Decision Letter 1]

2 Jan 2024

Growth patterns of children under 5 years old in Rural area northern of Abha, Aseer Region, Saudi Arabia

PONE-D-23-32367R1

Dear Dr. Safar Abadi Alsaleem,

Thank you for submitting your manuscript to PLOS ONE. We’re pleased to inform you that your manuscript has been judged scientifically suitable for publication and will be formally accepted for publication once it meets all outstanding technical requirements.The author has addressed all the comments raised by the reviewer. However, the author should revise and format the manuscript correctly after thoroughly evaluating the reviewer's comments and my assessment. 

Kind regards,

Satish Rojekar, Ph.D.

Academic Editor

PLOS ONE

Additional Editor Comments (optional):

Thank you for submitting your manuscript to PLOS ONE. The author has addressed all the comments raised by the reviewer. However, the author should revise and format the manuscript correctly after thoroughly evaluating the reviewer's comments and my assessment. I recommend accepting the manuscript with minor corrections.

Reviewers' comments:

Reviewer's Responses to Questions

**Comments to the Author**

1. If the authors have adequately addressed your comments raised in a previous round of review and you feel that this manuscript is now acceptable for publication, you may indicate that here to bypass the “Comments to the Author” section, enter your conflict of interest statement in the “Confidential to Editor” section, and submit your "Accept" recommendation.

Reviewer #1: All comments have been addressed

Reviewer #2: All comments have been addressed

2. Is the manuscript technically sound, and do the data support the conclusions?

Reviewer #1: Yes

Reviewer #2: Yes

3. Has the statistical analysis been performed appropriately and rigorously? 

Reviewer #1: Yes

Reviewer #2: Yes

4. Have the authors made all data underlying the findings in their manuscript fully available?

Reviewer #1: Yes

Reviewer #2: Yes

5. Is the manuscript presented in an intelligible fashion and written in standard English?

Reviewer #1: Yes

Reviewer #2: Yes

6. Review Comments to the Author

Reviewer #1: Following are the comments for the author

1. Please make the font style and size consistent as per the Journal guidelines.

2. Check the whole manuscript for formatting errors and please remove the track changes and comments from the word document.

3. Please make the formatting for the attached questionnaire. Please correct the spelling of "exclusive" in question 8.b.

Reviewer #2: All the comments are addressed by the authors. As of now, there is no any further comment from the reviewer.

7. PLOS authors have the option to publish the peer review history of their article (what does this mean?). If published, this will include your full peer review and any attached files.

Reviewer #1: No

Reviewer #2: No

---

## [Editor Report · Acceptance letter]

12 Feb 2024

PONE-D-23-32367R1 

PLOS ONE

Dear Dr. Alsaleem, 

I'm pleased to inform you that your manuscript has been deemed suitable for publication in PLOS ONE. Congratulations! Your manuscript is now being handed over to our production team.

Kind regards, 

on behalf of

Dr. Satish Rojekar 

Academic Editor

PLOS ONE